AI-SPedia: a novel ontology to evaluate the impact of research in the field of artificial intelligence

Maatouk Yasser ymaatouk@kau.edu.sa
Faculty of Computing and Information Technology, King Abdulaziz University , Jeddah , Saudi Arabia
Ashraf Imran
Electronic publication date: 2022 Sep 22
Publication date: 2022
Volume: 8
Electronic Location ID: e1099
Received 2022 Jun 29; Accepted 2022 Aug 22
Copyright: ©2022 Maatouk
Copyright year: 2022
Copyright holder: Maatouk
License: This is an open access article distributed under the terms of the Creative Commons Attribution License, which permits unrestricted use, distribution, reproduction and adaptation in any medium and for any purpose provided that it is properly attributed. For attribution, the original author(s), title, publication source (PeerJ Computer Science) and either DOI or URL of the article must be cited.
License URL: https://creativecommons.org/licenses/by/4.0/

Keywords: Altmetrics, Bibliometrics, Semantic web, Ontology

Funding: Deanship of Scientific Research (DSR), King Abdulaziz University, Jeddah DF-771-611-1441 This project was funded by the Deanship of Scientific Research (DSR), King Abdulaziz University, Jeddah, under Grant No. DF-771-611-1441. The funders had no role in study design, data collection and analysis, decision to publish, or preparation of the manuscript.

==============================
Background

Sharing knowledge such as resources, research results, and scholarly documents, is of key importance to improving collaboration between researchers worldwide. Research results from the field of artificial intelligence (AI) are vital to share because of the extensive applicability of AI to several other fields of research. This has led to a significant increase in the number of AI publications over the past decade. The metadata of AI publications, including bibliometrics and altmetrics indicators, can be accessed by searching familiar bibliographical databases such as Web of Science (WoS), which enables the impact of research to be evaluated and identify rising researchers and trending topics in the field of AI.

Problem description

In general, bibliographical databases have two limitations in terms of the type and form of metadata we aim to improve. First, most bibliographical databases, such as WoS, are more concerned with bibliometric indicators and do not offer a wide range of altmetric indicators to complement traditional bibliometric indicators. Second, the traditional format in which data is downloaded from bibliographical databases limits users to keyword-based searches without considering the semantics of the data.

Proposed solution

To overcome these limitations, we developed a repository, named AI-SPedia. The repository contains semantic knowledge of scientific publications concerned with AI and considers both the bibliometric and altmetric indicators. Moreover, it uses semantic web technology to produce and store data to enable semantic-based searches. Furthermore, we devised related competency questions to be answered by posing smart queries against the AI-SPedia datasets.

Results

The results revealed that AI-SPedia can evaluate the impact of AI research by exploiting knowledge that is not explicitly mentioned but extracted using the power of semantics. Moreover, a simple analysis was performed based on the answered questions to help make research policy decisions in the AI domain. The end product, AI-SPedia, is considered the first attempt to evaluate the impacts of AI scientific publications using both bibliometric and altmetric indicators and the power of semantic web technology.

Introduction

Experimental results and research may be published and shared within the scientific community in various types of documents, such as journal articles. The metadata of these scholarly documents can be used to identify articles in the same domain. This is expected to increase cooperation by identifying researchers with common interests. Artificial intelligence (AI) is one of the most important research areas and has found extensive application in several fields. AI has become increasingly popular in recent years because of its ability to solve complex practical problems and integrate different systems. Consequently, research pertaining to AI has increased in the past 20 years. Academia and industry expect research interest in the AI field to increase in the future. Moreover, according to many bibliographical databases, such as Web of Science (WoS), the number of AI publications has clearly risen in the past decade (Lei & Liu, 2019; Mellit & Kalogirou, 2008; Web of Science, 2021), as shown in Fig. 1.

WoS is one of the most important bibliographical databases in that it provides a gateway through which to search for and access the metadata of scholarly articles in all fields of research, including the AI domain. The structure and presentation of content in the WoS database provide useful information on the articles, in addition to the more common concern with the most important traditional measurements, such as bibliometrics (Web of Science, 2021). Bibliometrics are used to perform statistical analyses to evaluate research impact and determine the importance of a specific scientific publication. Bibliometrics can reveal the temporal evolution of research on a topic, using the number of citations as an example. Additionally, it can be applied to evaluate authors’ scientific output, such as their h-index indicator, or to evaluate journals in certain fields, such as their journal impact factor (JIF) (Archambault et al., 2009; Durieux & Gevenois, 2010; Franceschet, 2010).

We can improve the evaluation of the impact of a research article or researcher in a specific field using the WoS database. One limitation of bibliographical databases, including WoS, is that they do not consider the altmetric indicators of scientific documents, similar to the way in which they consider bibliometrics. Instead, these databases evaluate the impact of scientific research or researchers without considering these valuable altmetric parameters.

Altmetric measurements can capture the attention of online platforms and reflect trends in laypeople’s opinions. Furthermore, these measurements are considered to be metrics in the form of qualitative data, and thus complementary to traditional bibliometrics. Altmetric data sources include online platforms such as the following:

Figure 1 Annual variation in the number of publications concerned with AI based on WoS database.

– Public policy documents

– Peer reviews on Faculty of 1,000

– Citations in open online encyclopedias such as Wikipedia

– Exchange of views on blogs related to research

– Media coverage such as YouTube

– Reference managers such as Mendeley

– Mentions on social media networks such as Facebook

Overall, altmetrics include references to scholarly work on many online sources on the Web that clarify and explain how specific scholarly documents are discussed and used around the world (Altmetric, 2021).

The altmetric attention score (AAS) and donut shape were designed to identify the extent and type of attention received by a specific scholarly document alongside bibliometric indicators. AAS, which is a weighted count, is derived using an automated algorithm to represent the amount of attention attracted by a research output. The importance of this weighted count scenario is that it reflects the relative reach of all types of altmetric source. For example, an average newspaper article is weighted more heavily than a simple tweet because it attracts greater attention to research output (Altmetric, 2021).

Another way to improve a bibliographical database is to enhance the format in which the metadata of scientific documents are presented. Currently, these huge databases of scholarly articles provide publication metadata in a format that is understandable by humans yet not by machines for processing and answering smart queries. This traditional format limits users to keyword-based search.

Several studies have been conducted to extract structured information from scientific documents to gather semantically enriched data from existing sources. For example, DBLP (Aleman-Meza et al., 2007; DBLP, 2021), SPedia (Ahtisham, 2018; Ahtisham & Aljohani, 2016; Aslam & Aljohani, 2020), VIVO (Corson-Rikert & Cramer, 2010), CERIF (Nogales, Sicilia & Jörg, 2014), Sapientia (Daraio et al., 2016), PharmSci (Say et al., 2020), and several other studies involving ontologies that translate data from existing data sources related to scientific research into a unified resource description framework (RDF). Consequently, end users can query the dataset to extract useful knowledge using SPARQL. The main limitation of these works is that they cover and focus only on the bibliometric side of research indicators and ignore most of the altmetric indicators, which are considered to be useful and complementary to traditional bibliometrics.

To enhance the data representation and overcome the limitations of bibliographical databases, we developed a semantically-enriched knowledge base, referred to as AI-SPedia, as an extension to SPedia (Ahtisham, 2018; Ahtisham & Aljohani, 2016) to include information about scientific publications and researchers in the field of AI. AI-SPedia differs from SPedia in the following two ways: (i) it uses bibliometric and altmetric data of scientific publications to evaluate the research impact of a specific document or researcher; and (ii) the bibliometric and altmetric data of scientific publications are extracted from various sources, and they are presented as semantically enriched data (i.e., RDF datasets). Thus, AI-SPedia can be used to submit smart queries using the SPARQL protocol, thereby eliminating the need to manually analyze thousands of records. The output of these queries can be used to perform multi-purpose analyses to facilitate decision making for research strategies and policies. In this context, the contributions of our study can be summarized as follows:

– The AI-SPedia contains structured information for AI publications that are available in WoS. All relational data were triplicated, primarily between authors, publications, and journals. The end product contains valuable information regarding AI publications.

– Producing semantically enriched datasets as RDF unified triples from two sources (bibliometrics and altmetrics) extracted from detailed information (e.g., the digital object identifier (DOI) of the article, h-index of the author, journal impact factor (JIF), altmetric score) of AI scientific publications.

– Interlinking RDF datasets by combining bibliometric and metric sources. The AI-SPedia may be interlinked to existing external open datasets, such as DBPedia, SPedia, or other related ontologies in the web of data.

– Checking the availability and reusability of AI-SPedia ontology by creating a list of competency questions that the AI-SPedia should be able to answer. These questions were designed to consider the end users of the final product.

– Querying the final datasets using the AI-SPedia SPARQL Endpoint enables users to ask smart and complex queries to the RDF datasets of AI-SPedia.

– Presenting a basic analysis based on the answers to obtain more knowledge and useful informatics charts that are helpful for decision makers.

The remainder of this paper is arranged as follows: a discussion of related work is followed by a description of the data sources and extraction of structured information. Then, AI-SPedia is described and the reasoning features presented. The next section contains a demonstration of the validation of AI-SPedia using various queries, after which the differences between AI-SPedia and bibliographical databases (i.e., the WoS database) are discussed. The final section concludes the paper and presents guidelines for future work.

Related Work

Several studies have been conducted that make use of bibliometric data to calculate the impact of a research article or the index of a researcher or author. At the same time, many researchers have proposed solutions and scientific approaches to extract structured information related to scientific documents to build semantically enriched data from existing sources to enable users to submit smart queries to semantically enriched data. Work related to indexing based on bibliometric data and querying based on semantic data is discussed below.

SwetoDblp (Aleman-Meza et al., 2007) contains semantically enriched data built from an existing source, DBLP (2021), which is a database that provides bibliographic information on publications in computer science. Its aim is to support researchers by providing free access to metadata and links to electronic editions of computer science publications. It contains valuable data on 5.5 million publications by more than 2.5 million authors. It includes information about more than 5,000 conferences and 1,700 journals (DBLP, 2021). To produce the RDF, Aleman-Meza et al. (2007) built an ontology, SwetoDblp, using an SAX-parsing process that performed transformations on a large XML file that exists on the DBLP. The proposed schema adopts many concepts and their relationships with the FOAF and Dublin core ontologies. Additionally, the OWL vocabulary was used to show the equivalence of classes and relations. The additional datasets facilitated the integration of many entities and their relationships in SwetoDblp. The final ontology was enriched by including data from other sources. This ontology has a large volume of real data about individuals, supporting a comprehensive search at a single location of publications by individuals involved in various research outputs for information on authors and co-authors, conferences, journals, information about editors, and venues (Aleman-Meza et al., 2007).

Similarly, SPedia (Ahtisham & Aljohani, 2016) is a knowledge base that provides semantically enriched metadata for scientific documents published by Springer. It contains information on approximately nine million scholarly documents using SpringerLink as its source. The goal of SPedia is to facilitate a semantic-based search to allow users to pose smart and complex queries and overcome the limitations of traditional keyword-based searching by creating a SPARQL endpoint to query the dataset (Ahtisham & Aljohani, 2016). The authors demonstrated how SPedia can be used to analyze trends in joint publications and research collaborations (Ahtisham, 2018). The results retrieved from the SPedia knowledge base using SPedia SPARQL Endpoint enables the performance of both individual researchers and organizations to be analyzed. This process can help decision makers set future research directions (Ahtisham, 2018). Moreover, SPedia is the first step toward achieving an LOD cloud of scholarly documents by well-known publishers such as Springer. When SPedia is interlinked to LOD, it can be used for improved collaboration and knowledge sharing among scientific authors. It can act as a central hub for linked open scientific publication data by linking to the LOD of scientific work by other publishers (Aslam & Aljohani, 2020). One major limitation of SPedia is that it does not consider altmetric data and is entirely dependent on bibliometric parameters.

Another effort attempted to combine the three primary sources of research information. The authors integrated these sources and exported CERIF and VIVO models (Nogales, Sicilia & Jörg, 2014). VIVO (Corson-Rikert & Cramer, 2010) is a semantically enriched network of researchers such as investigators, students, and technical staff. A bibliographic database of more than 7 million records was combined with data from Google Scholar. The information was saved in VIVO instances and the VIVO model was translated into CERIF using a transformation method that maps both data models (Corson-Rikert & Cramer, 2010; Nogales, Sicilia & Jörg, 2014).

Daraio et al. (2016) proposed a data management method by building an ontology. This work was the first to distinguish, preserve, and merge the scientific, technological, and innovation (STI) data necessary for policy-making. Sapientia was created as an ontology for the evaluation of multidimensional research. This provided a transparent platform for the assessment model. The authors of this approach clarified that an easy-access method for scientific document data can allow for an improved understanding of science (Daraio et al., 2016).

PharmSci (Say et al., 2020) is an effort to organize publication data in the pharmaceutical field. Its purpose is to facilitate knowledge discovery through effective ontology-based data integration by rendering data easier to access and reuse. The PharmSci ontology can be used by end users to identify trusted reference material in sufficient detail for experiments and procedures. The authors followed an ontological engineering development approach. They presented reasoning and inference-based techniques in their work to enhance the quality of data integration and derive new facts. This approach acts as an agreed upon model of a specific domain and provides machine-interpretable information for the knowledge discovery process. It produced successful results and provided an ontology that was ready for implementation in applications (Say et al., 2020).

Recently, Samuel & König-Ries (2022) presented a collaborative framework for the management of scientific experiments in academic publications named CAESAR, which represents a collaborative environment for scientific analysis with reproducibility. It allows scientists to capture, manage, query, and visualize the complete path of scientific experiments by including both types of data: computational and non-computational. CAESAR integrates the REPRODUCE-ME ontology to represent a holistic view of an experiment, describing the path it followed from its design to its result. The proposed framework was applied and tested on research projects in the microscopic area. The final product is anticipated to assist the scientific community to track the complete path of the provenance of the results described in scientific publications (Samuel & König-Ries, 2022).

Our work differs from the aforementioned approaches and is summarized as follows:

1. To the best of the author’s knowledge, AI-SPedia is the first attempt to combine bibliometric and altmetric indicators. As an example, it provides indexes of AI publications based not only on citations but also on their popularity on social media.

2. AI-SPedia includes semantically-enriched knowledge based only on publications in the AI field. The main source of information was the WoS database.

3. AI-SPedia can accept sophisticated queries through its SPARQL Endpoint, a feature that is not offered by a regular bibliographical database. The results can be used to evaluate the research impact of publications and authors in the AI domain by using both bibliometric and altmetric indicators.

Data sources (Bibliometrics and Altmetrics)

This study uses a combination of two types of data as its sources: bibliometrics and altmetrics. In this section, we describe the approach we followed to process, extract, and produce the data from several sources in RDF.

Bibliometrics is one of the types of data and measurements used to evaluate the research impact. It is mainly used to provide evidence of the impact of specific research outputs. Bibliometrics comprises several types of indicators, such as the number of citations of a scientific article, an author’s h-index, and the JIF. The first is the number of appearances of a specific research paper in the reference list of other scholarly documents, whereas the h-index is designed to measure author’s productivity. Most bibliometric indicators can be found in bibliographical databases such as WoS and Scopus. In this study, we used the WoS database as the source of bibliometric data. This database is one of the most important bibliographical databases, which dates back to 1900, and contains valuable information on publications from the sciences, social sciences, arts, and humanities. WoS has been used to extract bibliometric information about AI publications (Butler et al., 2017).

The evolution of social media has accelerated and changed the way in which knowledge is shared. The need for alternative metrics to evaluate research impacts has arisen because of the explosion in the number of scientific websites and social media networks. For example, ResearchGate and SpringerLink are familiar sites that provide information on academic publishing achievements. Twitter is an online social media platform that is considered a major source of announcements related to new scientific publications. The platform is widely used to share information and disseminate scholarly documents (Hassan et al., 2017). Owing to the increasing use of online social media, academic researchers are adopting and using these platforms (Piwowar, 2013).

The second source of data for this research work is altmetrics, a term coined by Priem and Hemminger in their 2010 research (Priem & Hemminger, 2010). Altmetrics is a data metric that captures and measures the attention received by individual articles or researchers on several online social media platforms. Altmetrics may include mentions of types of scientific publication content, such as the datasets used by authors or the communities in which the data were collected (Haustein et al., 2016; Thelwall & Nevill, 2018; Yu et al., 2017).

The main goal of altmetrics is to evaluate the research impact of scholarly materials. Thus, the AAS can be calculated from the number of times an article is mentioned, downloaded, or shared on social media, in blogs, or in newspapers. Several sources of altmetric data are available, such as PlumX, Impact Story, and PLOS/Lagotto; however, the altmetric data used in this study were sourced from altmetric.com (Ortega, 2020).

Altmetrics should be considered as complementary to traditional bibliometric measures. They can provide an overview of the interest in a particular research paper or topic and the discussions related to the topic. Moreover, altmetrics can provide an immediate signal of the effect of a specific paper, whereas citations need to accumulate over the years. As a result, one of the main objectives of this research is to combine data from both of these sources into a single knowledge base to widen the range of indicators.

Extracting structured information (Methodology)

One of the important issues that arise when working with data is information extraction. The extraction of information from these two data sources was an essential part of this research. Databases such as those containing altmetrics and bibliometrics are used in a structured format. Extracting information from data sources without altering the meaning is difficult. Semantic web (SW) technology is a solution to this challenge, as it can preserve the meaning of data from several types of sources. Consequently, machines can interpret and understand these data using RDF. RDF is used to add semantic meaning and provide a hierarchy of classes and properties (Gandhi & Madia, 2016; Maatouk, 2021).

Publications in the AI field were chosen as the focus of this study for reasons discussed earlier. Extraction was performed mainly on the two sources to obtain information on bibliometric and altmetric indicators using the LOPDF framework. Mapping was performed between the two sources of publication data in RDF, where each data item is triplified in the subject, predicate, and object formats. Figure 2 presents an overview of the AI-SPedia knowledge extraction process, which is used to extract data from data sources using the LOPDF framework (Aslam, 2021).

Figure 2 Overview of the AI-SPedia extraction process.

The altmetric data were in JSON format. Using Python script, we moved all the files into MongoDB software. Then, we downloaded the CSV files of all the AI publications from the WoS database. Next, the DOI was used to pair the AI publications that occur in both of the two data sources to obtain the final dataset. All data were imported in CSV file format. Then, the data were triplified using Protégé (which has the created ontology) to create an RDF model. The data were linked and loaded into the AI-SPedia.

– Altmetric Source: Data from altmetric.com were received in JSON format. Using Python script, we moved all the files into MongoDB, software that can hold large JSON files, such that we could query the data easily. For the purpose of this study, we extracted the main attributes defining an AI publication and its research impact, such as the DOI (as the primary key), the title of the publication, the name of the journal in which the article was published, its altmetric score, and other platform scores. In this study, we focused on Twitter, patents, news, Facebook, Google+, Wikipedia, and blogs scores, because these sources are responsible for generating more than 98% of all altmetric data. Patents and news were assigned larger weights than the other sources. Twitter alone accounted for approximately 75% of all altmetric data.

– Bibliometric Source: Many sources of bibliometric data are available. The most important bibliographical databases are WoS and Scopus. Several studies have discussed the features of each of these databases. In this study, we used WoS as the source of bibliometric data. Based on the journal category, we downloaded the CSV files of all the publications concerned with AI. The downloaded files also include the citation counts of all AI publications, the JIF for each AI journal and, for all AI publications, the first author’s name and h-index.

– Matching Process: Because DOIs are widely used to identify academic publications, we used the DOI to pair the AI publications that appear in both data sources (altmetrics and bibliometrics). The DOIs are used to uniquely identify both objects and documents. After the matching process, we obtained a final dataset of approximately 8,000 publications in the AI field that were also listed in the altmetric database (Data S1). All data were converted into CSV file format before they were imported to facilitate processing. We included all AI publications to build the first version of AI-SPedia for our case study.

– Triplifier Process: We started to triplify the data to create an RDF model. To achieve this, using Protégé, we first built an ontology with classes, properties, and relations between entities. In the next step, all the data extracted from bibliometric and metric sources were cleaned and temporarily saved in CSV files to map to the classes of the ontology. These data entities are linked to each other by various object and data-type properties to preserve the semantics and meaning of each instance. In each case, the data were published in RDF (as. nt files), and loaded into the AI-SPedia repository. The generated dataset was processed to be producible in open format such that the data could be linked to other open datasets and processed by machine.

Description of AI-SPedia and its building process

An ontology, in its basic definition, is a formal explicit description of knowledge as a set of concepts within a specific domain, and the relationships that link them. According to this definition, the creation of a knowledge base graph requires us to formally specify entities such as classes, individuals that represent instances of classes, the properties of each concept describing its various features and attributes, relations between the concepts, and the restrictions, rules, and axioms (Guarino, Oberle & Staab, 2009).

Classes are the focus of ontology. They describe the concepts of the selected domain, whereas the properties describe the classes and their instances. There are two types of properties: object and data. Both help us to relate entities and transform data into knowledge. Object properties link individuals to each other, and data properties link individuals to literal values, such as integers or float numbers. These properties may have several facets that describe the allowed value type. The classes that the property describes are referred to as the domain of the property, whereas the range defines the allowed classes of a property. In practice, as an incremental and iterative process, ontology development follows agile methodology (Guarino, Oberle & Staab, 2009).

To develop AI-SPedia, we identified the classes in the selected domain. Subsequently, we defined the properties and described their allowed values. Ultimately, the AI-SPedia ontology was reviewed and enhanced such that it can be used as a schema to accommodate data from both bibliometric and altmetric sources. Classes and both types of properties were added to the semantically enriched meanings of these properties. In the last step, a knowledge base graph was created by defining individuals (instances of the classes) and filling the value information of every property in addition to value constraints and restrictions. We also defined object properties with different reasoning features, such as inverse, functional, and transitive. Several data properties were also defined to link instances to literal values. Moreover, both the rdfs:domain and rdfs:range were defined for each property. After identifying the entities related to our data, we determined the relationships between the entities in our datasets.

The most important classes of our AI-SPedia ontology are AI-SPedia:Paper, AI-SPedia:Author, and AI-SPedia:Journal, as shown in Fig. 3. The main class in AI-SPedia is “Paper,” which represents publications in the AI field from the WoS database and altmetrics sources. The two most important properties by which to identify an AI research paper are its DOI and title, which are considered the main outputs when querying the AI-SPedia dataset using SPARQL.

Figure 3 Conceptual basic building blocks of the AI-SPedia ontology.

The other two classes that can interlink with the “Paper” class are “Author” and “Journal”:

1. Author Class represents the author of the AI articles. It has a propriety “h-index.’ The h-index is a numerical indicator of both productivity and the impact on a particular researcher. It can be calculated by considering both the number of publications and citation count. In addition, class “Author” has an object property “write.” The range of the property “write” is an instance of the “paper” class. The “write” property has an inverse property called “written_by.”

2. Journal Class represents the journal that publishes AI articles. It has data propriety “impact_factor.” Bibliometric IF is a scientometric indicator that can be used to measure the importance of an academic journal in a certain field. Usually, journals with higher IFs are more important than those with lower IFs. The IF of a journal can be calculated by dividing the number of citations received by the number of papers in the journal. The class, “Journal,” has an object property “publish.” The range of this property is an instance of the “Paper” class. The “publish” property has an inverse property called “published_in.”

Furthermore, our ontology includes several properties listed in Table 1. The “Paper” class has two important properties:

Table 1 Most important properties of AI-SPedia.

Property	Type	Domain	Range	Algebraic expression	
write	object	Author	Paper	P <write, Author, Paper>	
written_by	object	Paper	Author	P <written_by, Paper, Author>	
publish	object	Journal	Paper	P <publish, Journal, Paper>	
published_in	object	Paper	Journal	P <published_in, Paper, Journal>	
DOI	data	Paper	value	P <DOI, Paper, value>	
citation_number	data	Paper	value	P <citation_number, Paper, value>	
altmetrics_score	data	Paper	value	P <altmetrics_score, Paper, value>	
H_index	data	Author	value	P <H_index, Author, value>	
impact_factor	data	Journal	value	P <impact_factor, Journal, value>	
twitter_score	data	Paper	value	P <twitter_score, Paper, value>	
patent_score	data	Paper	value	P <patent_score, Paper, value>	
news_score	data	Paper	value	P <news_score, Paper, value>	

1. Citation Count. A citation is a reference to the source of information used in research. It simply informs readers that certain content in the work are from another source and gives credit to the original author. The citation impact indicator plays a significant role in the evaluation of research. Therefore, it has received considerable attention in scientometric studies. The main databases from which the citation counts were obtained were WoS and Scopus.

2. Altmetric Score. Altmetric sources may incorporate article views, downloads, or social media mentions. It is a more informative, readily available, article-level metric that can be used in addition to the citation count. Several studies have demonstrated a weak correlation between the citation count and altmetric score; however, they are complementary to each other and could be used as such. New publications may receive increased online attention directly after they are published, whereas bibliometric citations take longer to accumulate. This is because new publications are disseminated across the media and social networks. Sensationalism may arise in certain scientific documents, which may not be borne out by an academic setting.

Altmetrics includes more than a single online resource. In other words, several attributes can be used to compute the altmetric score; however, the most important online resources, which represent more than 98% of the altmetrics data, are Twitter, patents, news, Facebook, Google+, Wikipedia, and blogs. In AI-SPedia, we represent these resources as twitter_score, patent_score, news_score, Facebook _score, Google _score, Wikipedia _score, and blogs _score. To convert our datasets from the CSV format of an Excel sheet to RDF/OWL, we generated our own ontology and RDF triples using the ontology development tool Protégé, as described earlier during the extraction process. Figure 4 presents screenshots of the Protégé tool.

Figure 4 Snapshot of AI-SPedia ontology from Protégé software.

The ontology includes: (A) classes, (B) object properties, (C) data properties.

Machine reasoning, as part of AI-SPedia, is a key feature of semantic technologies that differentiates RDF data representation from the traditional relational database management systems (RDBMS). In particular, machine reasoning allows us to set different rules to help machines infer new knowledge that is not explicitly mentioned. In AI-SPedia, we use reasoning by defining rules to drive new logical facts. For example, the following rules were applied to export new axioms and declarations to instances in AI-SPedia.

The first rule states that for every AI paper in AI-SPedia, there exists an author who wrote the paper and an AI journal in which the paper was published. For example, AI-SPedia does not contain any papers without the name of the first author or the journal in which the paper was published. (1) ∀?paper→∃written_by?paper,?author∧published_in?paper,?journal

The second rule states that if the AI paper has a score for at least one of the altmetric parameters, we can infer that this AI paper has an altmetric score. (2) ∀twitter_score?paper,float_number→∃altmetrics_score?paper,float_number

The third rule states that, for any paper that has been cited at least once, it can be inferred that there exists an h-index indicator for the author of this paper and an IF for the journal that published the paper. (3) ∀citation?paper,integernumber→∃h−index?author,integer_number∧impact_factor?journal,float_number.

This clearly indicates that the definition of rules can be important and useful by enabling machines to reason and derive new facts that are not explicitly expressed. This feature of the semantic web is considered to be the major difference that sets it apart from a traditional RDBMS by offering improved RDF data representation.

Results of Querying AI-SPedia

In this section, we demonstrate the validity and reusability of the AI-SPedia datasets. A list of competency questions was created to validate the accuracy of the AI-SPedia dataset and its semantic model. All questions derived from the content of AI-SPedia. The aim was to check whether the AI-SPedia repository could correctly answer the questions, and to validate the correctness of the ontology in the context in which it is used. The competency questions were designed according to the end users of the final product, probably academic researchers interested in publications in the field of AI or researchers who could benefit from this ontology schema and the corresponding repositories. The list of competency questions is as follows:

1. List of AI publications with their citation numbers and altmetric scores.

2. List of AI publications that have patent scores.

3. List of AI publications mentioned in news.

4. List of AI publications that are mentioned extensively on Twitter.

5. List of AI publications that contain Facebook, Google+, Wikipedia, and blogs scores.

The results of these questions assert that the designed model includes sufficient details of AI research (mainly on bibliometrics and altmetrics indicators). We implemented SPARQL queries for each defined question. All queries regarding the questions and possible answers were written. The results revealed that the structure and content of AI-SPedia are valid because the competency questions are answered and validated correctly in response to the SPARQL queries. Furthermore, we analyzed the answers to the questions to extract knowledge and useful information, according to the process shown in Fig. 5.

Q1: List of AI publications with their number of citations and altmetric scores

The AI-SPedia SPARQL endpoint can also be used to query AI publications based on their altmetric scores and citation counts. The code below shows a query executed through the SPARQL endpoint, and Table 2 presents part of the query results. These results can be used to analyze and compare altmetrics and bibliometric indicators. For example, Table 2 summarizes the DOI, citation count, and altmetric score. These results could be analyzed to show the correlation between the citation count and altmetric score because the correlation coefficient is useful to determine and understand the relationship between the two variables. The Pearson correlation coefficient of the two sets of variables was calculated to have a value of 0.2. This value is close to zero, which indicates that there is no correlation between the citation count and metric score. A weak correlation between altmetrics and bibliometrics is to be expected, similar to previous studies in other research areas such as orthopedics (Collins et al., 2021), anesthesiology (Rong et al., 2020), and implantology (Warren, Patel & Boyd, 2019).

Figure 5 Working process to answer the competency questions.

Table 2 Snapshot of the AI publication list with their citations and altmetric scores.

DOI	Citations	Altmetric score	
10.1016/j.media.2016.10.004	501	30.9	
10.1016/j.media.2009.05.004	683	9	
10.1016/j.inffus.2006.10.002	123	6	
10.1016/j.patrec.2014.02.021	5	107.25	
10.1007/s11263-007-0090-8	1001	3	
10.4018/jswis.2009081901	1539	28.894	
10.1007/s10849-015-9213-8	1	0.25	
10.1049/iet-bmt.2014.0040	24	66.58	
10.1016/j.neucom.2007.12.038	48	3	

SELECT ?DOI ?paper ?citation_number ?altmetrics_score

WHERE { ?paper ont:DOI ?DOI .

? Paper onts:citation _number citation_number.

?paper ont:altmetrics_score ?altmetrics_score .}

The results we obtained in response to Q1 were additionally analyzed to extract a list of the top 15 publications with (1) the highest metric scores (Table 3) and (2) the most citations (Table 4). These two lists had only one paper in common: a review paper entitled “Deep learning in neural networks.” This paper is considered a historical survey that succinctly summarizes relevant and recent works in deep artificial neural networks, pattern recognition, and machine learning. The author cited more than 1,000 references, which were consulted during the survey.

Table 3 Top 15 AI papers based on their altmetric score.

Title	Citations	Altmetric_Score	
Deep Learning in Neural Networks: An Overview	3353	481	
Brainprint: Assessing the uniqueness, collectability and permanence of a novel method for ERP biometrics	53	363.394	
Playing Counter-Strike versus running: The impact of leisure time activities and cortisol on intermediate-term memory in male students	1	196.8	
Information systems and task demand: An exploratory pupillometry study of computerized decision-making	9	179.5	
Turing learning: A metric-free approach to inferring behavior and its application to swarms	6	177.646	
Unsupervised real-time anomaly detection for streaming data	62	134.15	
Gradient boosting machines –A tutorial	189	114.63	
Should I send this message? Understanding the impact of interruptions, social hierarchy and perceived task complexity on user performance and perceived workload	32	114.58	
Computer analysis of similarities between albums in popular music	5	107.25	
Integrated local binary pattern texture features for classification of breast tissue imaged by optical coherence microscopy	15	101.43	
Systematic evaluation of convolution neural network advances on the ImageNet	21	98.4	
Complexity of n-Queens completion	6	97.61	
Detecting criminal organizations in mobile phone networks	55	95.226	
Predicting crime using Twitter and kernel density estimation	173	86.1	
Permanence of the CEREBRE brain biometric protocol	7	82.696	

Table 4 Top 15 AI papers based on their citation count.

Title	Citations	Altmetric_ Score	
LIBSVM: A library for support vector machines	15721	29.644	
Speeded-Up Robust Features (SURF)	6282	21	
ImageNet large-scale visual recognition challenge	5041	28.85	
A fast iterative shrinkage-thresholding algorithm for linear inverse problems	4142	15	
Deep learning in neural networks: An overview	3353	481	
The Pascal Visual Object Classes (VOC) challenge	3231	9	
Data clustering: 50 years beyond K-means	2991	21.758	
The Split Bregman method for L1-regularized problems	2271	3	
Top 10 algorithms in data mining	1880	29.15	
Selective search for object recognition	1612	1.25	
Linked data - The story so far	1539	28.894	
A practical tutorial on the use of nonparametric statistical tests as a methodology for comparing evolutionary and swarm intelligence algorithms	1523	11.1	
A new alternating minimization algorithm for total variation image reconstruction	1032	6	
A survey on vision-based human action recognition	1025	6	
LabelMe: A database and web-based tool for image annotation	1001	3	

For a survey paper such as this, the considerable amount of information and sources would be expected to elevate the paper to the top of the list of both bibliometric and altmetric indicators. The journal Neural Networks, in which this paper is published, had a JIF of 5.785 at the time of publication, which is not considered particularly high among journals in this field, although it is in Q1 (top 25%) in the computer science field. The author of this paper has an h-index of 31, which is considered very good compared with other authors in the field.

Table 5 Snapshot of AI publications that have patent scores.

DOI	Title	
10.1016/j.inffus.2013.04.006	A_survey_of_multiple_classifier_systems_as_hybrid_systems	
10.1002/int.21521	On_Z-valuations_using_Zadeh’s_Z-numbers	
10.1016/j.inffus.2010.03.002	Performance_comparison_of_different_multi-resolution_transforms_for_image_fusion	
10.1016/j.media.2009.05.004	Statistical_shape_models_for_3D_medical_image_segmentation:_A_review	
10.1016/j.media.2009.07.011	A_review_of_3D_vessel_lumen_segmentation_techniques:_Models__features_and_extraction_schemes	

Q2: List of AI publications that have a patent score

The following SPARQL code fetches AI publications that have a patent score. Such publications are mentioned in the patent reference sections. Table 5 lists part of the query results. Patent scores can contribute to the AAS. Each patent is assigned to a jurisdiction. The jurisdiction is used by altmetric.com to decide the contribution to the final score. Researchers interested in AI papers that have a patent score can easily use this query to extract a list of such AI papers.

SELECT ?DOI ?paper ?patent_score

WHERE { ?paper ont:DOI ?DOI .

?paper ont:patent_score ?patent_score .

FILTER(?patent_score >0)}

Q3: List of AI publications mentioned in the news

The following SPARQL code fetches AI papers that were mentioned in the news. Table 6 presents part of the list. Clearly, not all papers were mentioned in the news. The news score is considered an important altmetric, and researchers interested in papers mentioned in the news can use this query to fetch a list of these papers. A news score greater than zero indicates that a paper is mentioned in the news.

Table 6 Snapshot of list of AI publications mentioned in the news.

DOI	Title	
10.1016/j.neucom.2015.04.025	Brainprint: Assessing the uniqueness collectability and permanence of a novel method for ERP biometrics	
10.1016/j.cogsys.2016.01.002	Playing Counter-Strike versus running: The impact of leisure time activities and cortisol on intermediate-term memory	
10.1007/s11721-016-0126-1	Turing learning: a metric-free approach to inferring behavior and its application to swarms	
10.1613/jair.5512	Complexity of n-Queens Completion	
10.1016/j.patcog.2016.09.027	Multifeature-based benchmark for cervical dysplasia classification evaluation	

SELECT ?DOI ?paper ?news_score

WHERE { ?paper ont:DOI ?DOI .

?paper ont:news_score ?news_score .

FILTER(?news_score >0)

Q4: List of AI publications with extensive twitter mentions

Certain studies have stimulated arguments and discussions on Twitter. Consequently, it would be interesting and useful to know more about these and why they are extensively mentioned on Twitter. The following SPARQL query fetches a list of AI papers that are mentioned extensively on Twitter based on their Twitter _score. Table 7 presents part of this list. Readers or researchers interested in knowing which AI papers gave rise to discussions and associated arguments in the AI field can use this query to fetch those with extensive mentions on Twitter. We assume that an AI paper with a Twitter score of over 50 has been mentioned extensively.

SELECT ?DOI ?paper ?twitter_score

WHERE { ?paper ont:DOI ?DOI .

? Paper ont: Twitter score ?twitter_score .

FILTER(?twitter_score >50)}

Q5: List of AI Publications that have Facebook, Google+, Wikipedia, and blog scores

The following SPARQL code fetches the AI papers mentioned on Facebook, Google+, in Wikipedia, and in blogs. Table 8 lists examples of the query results.

SELECT ?DOI ?paper ?facebook_score ?google_score ?wikipedia_score ? blog_score

WHERE { ?paper ont:DOI ?DOI .

?paper Ot:facebook_score ?facebook_score .

?paper ont:google_score ?google_score .

?paper Ot:wikipedia_score ?wikipedia_score .

? Paper ont:blogscore blog_score .}

The above-mentioned competency questions demonstrated the type of information that can be extracted from the AI-SPedia dataset. Bibliographical databases such as the WoS database are not able to provide answers to these questions because of the limitations discussed earlier. The next section provides a detailed discussion of the advantages of the AI-SPedia knowledge base compared with regular bibliographical databases.

Table 7 Snapshot of the list of AI publications that have extensive mentions on Twitter.

DOI	Title	Twitter Score	
10.1016/j.dss.2017.02.007	Information systems and task demand: An exploratory pupillometry study of computerized decision making	2375	
10.1016/j.neunet.2014.09.003	Deep learning in neural networks: An overview	486	
10.1016/j.eswa.2013.12.050	A multi-objective hyper-heuristic based on choice function	145	
10.1080/09540091.2016.1271400	Principles of robotics: regulating robots in the real world	142	
10.3389/fnbot.2013.00021	Gradient boosting machines a tutorial	134	

Table 8 Snapshot of the list of AI publications with a score in Facebook, Google+, Wikipedia and blogs.

DOI	Score	
10.1016/j.artint.2015.03.009	Facebook = 22	
10.1016/j.neunet.2014.09.003	Facebook = 21	
10.1016/j.neunet.2014.09.003	Google+ = 82	
10.1016/j.neunet.2012.09.016	Google+ = 27	
10.1016/j.neunet.2014.09.003	Wikipedia = 14	
10.1016/j.neunet.2011.06.014	Wikipedia = 5	
10.1016/j.neucom.2015.04.025	Blogs = 13	
10.1016/j.neunet.2014.09.003	Blogs = 7	

Discussion

This section describes the main differences between the AI-SPedia knowledge base and conventional bibliographical databases and highlights the role of AI-SPedia as the preferred source of publication metadata and information in the AI domain. A distinct advantage of AI-SPedia is its ability to evaluate the research impact of articles and author indices by including more than merely bibliographical data. More specifically, bibliographical databases such as WoS emphasize bibliometric indicators, such as the citation count, JIF and, authors’ h-index, whereas, in addition to bibliometric indicators, AI-SPedia contains the altmetric indicators. AI-SPedia therefore offers a more comprehensive collection of the most important metric scores.

Additionally, one of the most important features of AI-SPedia is its ability to infer new facts from existing ones and perform machine reasoning based on the created rules, which is not a feature of any database system. The use of reasoning and inference increases the power of expression and reduces the required knowledge base. AI-SPedia was designed with these features to enable new facts to be derived that are not explicitly expressed. We defined several rules to infer new logical meanings and discover new facts among instances, as discussed in the reasoning and inference sections.

Another respect in which AI-SPedia is superior is the method it uses for exporting data. Bibliographical databases commonly export data in a common file format, such as CSV, which accommodates no more than 1,000 records. In contrast, in AI-SPedia, RDF is used as the graph-based representation format for publication metadata. This facilitates querying of the data using the SPARQL protocol, enabling users to conduct a semantic-based smart search rather than being restricted to keyword-based searching.

The last major difference is the ability of AI-SPedia to interlink datasets. This allows links, which are understandable not only to humans but also to machines, to be shared and created between data entities from different sources. In this regard, LOD is one of the most important implementations of SW technologies applied to AI-SPedia to lay down the best practice for creating these links. In other words, the power of AI-SPedia lies in its ability to connect multiple sources. Consequently, the end user has the benefit of being able to access several datasets from existing disparate resources.

The only limitation of the current version of AI-SPedia is that it only contains AI publications, whereas bibliographical databases include publications from all areas of research, including AI. However, compared to previous studies, AI-SPedia is the first attempt to combine both bibliometric and altmetric indicators. As an example, it provides indexes of AI publications based not only on citations but also on their popularity on social media. Further, AI-SPedia can answer smart queries through its SPARQL Endpoint feature, whereas regular bibliographical databases do not have this capability. The results can be used to evaluate the research impact of publications and authors in the AI domain by using both bibliometric and altmetric indicators.

Conclusion and Future Work

The AI field has witnessed a sharp increase in the number of publications. Many bibliographical databases such as WoS provide comprehensive metadata on scientific documents, their impact, researchers, and their ranking. However, for several reasons, the metadata provided cannot be used to their full potential. A plausible reason for this limited use is that bibliographical databases evaluate the impact of a research article or the rank of a researcher only on the basis of traditional bibliometric indicators, because they do not contain a wide range of altmetric indicators. For example, they do not provide any information regarding non-specialist opinions on specific publications. Scientific documents can capture the attention of users of online platforms, such as Twitter and Facebook. References to documents on these platforms are considered metrics and these altmetrics can be used to complement traditional bibliometrics. Another shortcoming of existing bibliographical databases is that the availability of metadata for scientific documents in a specific format restricts users to keyword-based searching and does not enable implicit knowledge to be extracted from the available data.

To overcome these limitations, we considered both bibliometric and altmetric parameters to evaluate the research impact in addition to the researcher’s ranking. We also translated the collected data into RDF, which is mapped to AI-SPedia (an extended version of SPedia), to ultimately enable users to submit smart queries rather than only conducting keyword-based searches.

As a case study, we developed a structured, semantics-based repository named AI-SPedia, the first version of which includes semantically enriched data from approximately 8,000 AI publications. AI-SPedia covers all bibliometric indicators and the seven most important altmetric sources. The correctness of AI-SPedia was validated by creating a list of five competency questions AI-SPedia should be able to answer. Additionally, we implemented SPARQL queries for each question. The results revealed that AI-SPedia can evaluate research impact by exploiting knowledge that is not explicitly mentioned but could be extracted using the power of semantics. Furthermore, we analyzed the answers to the questions to determine whether they could facilitate decision-making and policy-making. According to this analysis, the correlation between bibliometrics and altmetrics is weak, indicating that these data sources are complimentary. The end product, AI-SPedia, is considered a first attempt to calculate indexes using both bibliometric and altmetric indicators for publications in AI using semantic web technology.

In future work, we plan to upgrade the AI-SPedia knowledge base to serve as a comprehensive recommender system with more detailed scientific indicators and parameters, including properties related to the body of scientific articles. We plan to increase the publications and data included in AI-SPedia by using the Scopus database, in addition to WoS. The next version of AI-SPedia, in addition to bibliometrics and altmetrics indicators, will include more attributes, such as the keywords listed by the author and the minor category within AI. Furthermore, a new indicator that is particularly suitable for the AI field should be created to convey a complete understanding to the reader and provide greater meaning to the value of AI publications. The same approach can be used for different trending research fields such as computer networks, cancer research, COVID19, and so on. I have chosen artificial intelligence domain for the reasons that are discussed earlier in the introduction section.

Supplemental Information

Supplemental Information 1 Raw data

Publications in artificial intelligence field: the DOI, authors names, journals, bibliometric and altmetric parameters.

Click here for additional data file.

I would like to thank the KAU library for providing resources and Editage for English language editing.

Additional Information and Declarations

Competing Interests

Author Contributions

Data Availability

The author declares there are no competing interests.

Yasser Maatouk conceived and designed the experiments, performed the experiments, analyzed the data, performed the computation work, prepared figures and/or tables, authored or reviewed drafts of the article, and approved the final draft.

The following information was supplied regarding data availability:

The raw data is available in the Supplementary File.

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
