# Peer review of "AI-SPedia: a novel ontology to evaluate the impact of research in the field of artificial intelligence"

_PeerJ Computer Science, doi:10.7717/peerj-cs.1099_

## Round 0.1 · original submission · Major Revisions

The authors are requested to make, "major revisions".

Reviewer 1 ·

Basic reporting

The author has done good work, Presentation is also good. There are some comments for the author?

What is the significance of this study? How you can differentiate it from literature?
You mentioned that "Moreover, an analysis was performed based on the answered questions to help make research decisions"

What kind of answered questions? are these contextual-based analyses?
is there any statistical approach to analyzing the significance of the results?
Difficult to understand the finding of the study.

Experimental design

No comments

Validity of the findings

No comments

Additional comments

No comments

Reviewer 2 ·

Basic reporting

The author should mention how the twitter_score, and patent_score are given to selected AI publications. Are these scores collected from external sources?

Experimental design

Please include the discussion on the applicability of the proposed approach in other field research articles. For instance, computer networks, cancer research, etc.

Validity of the findings

The scalability of the proposed approach should be discussed. For example, why was the article count of 8000 selected for evaluation.

---

## Round 0.2 · accepted · Accept

Based on the reviewers' suggestion, the paper is accepted.

Reviewer 1 ·

Basic reporting

The author gives an explanation for my all concerns, I have no other concerns.

Experimental design

no comments

Validity of the findings

no comments

Additional comments

no comments

Reviewer 2 ·

Basic reporting

no comment

Experimental design

no comment

Validity of the findings

no comment

Additional comments

All my questions were clarified.